# Vitronectin-derived bioactive peptide prevents spondyloarthritis by modulating Th17/Treg imbalance in mice with curdlan-induced spondyloarthritis

Hong Ki Min[1◉], JeongWon Choi[2◉], Seon-Yeong Lee[2], A. Ram Lee[2,3], Byung-Moo Min[4], Mi-La Cho[2,3,5‡]*, Sung-Hwan Park[2,6‡]*

1 Division of Rheumatology, Department of Internal Medicine, Konkuk University Medical Center, Seoul, Republic of Korea, 2 Rheumatism Research Center, Catholic Research Institute of Medical Science, The Catholic University of Korea, Seoul, Republic of Korea, 3 Department of Biomedicine & Health Sciences, College of Medicine, The Catholic University of Korea, Seoul, Republic of Korea, 4 Department of Oral Biochemistry and Program in Cancer and Developmental Biology, Dental Research Institute, Seoul National University School of Dentistry, Seoul, Republic of Korea, 5 Laboratory of Immune Network, Conversant Research Consortium in Immunologic disease, College of Medicine, The Catholic University of Korea, Seoul, Republic of Korea, 6 Division of Rheumatology, Department of Internal Medicine, College of Medicine, The Catholic University of Korea, Seoul, Republic of Korea

◉ These authors contributed equally to this work.
‡ MC and SP also contributed equally to this work.
* iammila@catholic.ac.kr (MC); rapark@catholic.ac.kr (SP)

**Data Availability Statement:** All relevant data are within the paper and its Supporting Information files.

## Abstract

### Purpose

Spondyloarthritis (SpA) is a systemic inflammatory arthritis mediated mainly by interleukin (IL)-17. The vitronectin-derived bioactive peptide, VnP-16, exerts an anti-osteoporotic effect via β1 and αvβ3 integrin signaling. SpA is associated with an increased risk of osteoporosis, and we investigated the effect of VnP-16 in mice with SpA.

### Methods

SpA was induced by curdlan in SKG ZAP-70[W163C] mice, which were treated with vehicle, celecoxib, VnP-16, or VnP-16+celecoxib. The clinical score, arthritis score, spondylitis score, and proinflammatory cytokine expression of the spine were evaluated by immunohistochemical staining. Type 17 helper T cell (Th17) and regulatory T cell (Treg) differentiation in the spleen was evaluated by flow cytometry and in the spine by confocal staining. Splenocyte expression of signal transducer and activator of transcription (STAT) 3 and pSTAT3 was evaluated by in vitro Western blotting.

### Results

The clinical score was significantly reduced in the VnP16+celecoxib group. The arthritis and spondylitis scores were significantly lower in the VnP-16 and VnP16+celecoxib groups than the vehicle group. In the spine, the levels of IL-1β, IL-6, tumor necrosis factor-α, and IL-17

**Funding:** This research was funded by a grant from the Korea Health Technology R&D Project through the Korea Health Industry Development Institute, funded by the Ministry of Health & Welfare of the Republic of Korea (grant number HI20C1496, grant receiver : SH Park). The funder had no role on the study design, data collection and analysis, decision to publish, or preparation of the manuscript.

**Competing interests:** The authors have declared that no competing interests exist.

expression were reduced and Th17/Treg imbalance was regulated in the VnP-16 alone and VnP-16+celecoxib groups. Flow cytometry of splenocytes showed increased polarization of Tregs in the VnP-16+celecoxib group. *In vitro*, VnP-16 suppressed *p*STAT3.

## Conclusions

VnP-16 plus celecoxib prevented SpA progression in a mouse model by regulating the Th17/Treg imbalance and suppressing the expression of proinflammatory cytokines.

## Introduction

Spondyloarthritis (SpA) is an inflammatory arthritis, which affects about 0.20 to 1.61% of the population [1]. SpA has two subtypes: axial SpA (axSpA) and peripheral SpA [2]. AxSpA induces axial joint inflammation and eventually new bone formation at the vertebral corner, also termed syndesmophyte formation [3]. The treatment of axSpA aims to reduce the inflammatory response and suppress abnormal bony bridging of axial joints [3]. The treatment guidelines for axSpA recommend non-steroidal anti-inflammatory drugs (NSAIDs) as the first-line therapy for patients with axSpA and active arthritis symptoms [4]. Interleukin (IL)-17 mediates inflammation and abnormal hyperosteosis in axSpA [5]. Various immune cells of patients with axSpA express IL-17 [6], and type 17 helper T (Th17) cells are one of the main cells expressing IL-17 in axSpA. Th17 cells are upregulated in the peripheral blood of patients with axSpA [7]. Furthermore, the circulating Th17 level and regulatory T cell (Treg) abundance are positively and negatively, respectively, correlated with axSpA disease activity [8]. Therefore, targeting the Th17/Treg imbalance may have therapeutic potential for axSpA. We reported that modulation of the Th17/Treg imbalance and suppression of pro-inflammatory cytokines by signal transducer and activator of transcription (STAT) 3 inhibitors, rebamipide and protein inhibitor of activated STAT3, prevented axSpA in mice [9, 10].

Osteoporosis is more common in patients with axSpA than in the general population [11, 12], and low mineral bone density can occur in the early stages of axSpA [13]. Patients with ankylosing spondylitis (AS) have a 7.1-fold increased risk of vertebral fracture than the general population, and that risk is elevated further in the presence of other inflammatory arthritis or autoimmune diseases [14]. The increased risk of osteoporosis and related fracture in axSpA is related not only to known risk factors for osteoporosis but also systemic inflammation [15]. Bisphosphonate is the most commonly used anti-osteoporotic agent and inhibits osteoclasts and monocyte/macrophage lineage cells [16]. The anti-inflammatory effect of bisphosphonate on monocyte lineage cells has been investigated [17]. An open-label pilot study of pamidronate pulse therapy showed modest efficacy for AS [18, 19]. Therefore, anti-osteoporotic agents could also exert an anti-inflammatory effect. Vitronectin-derived bioactive peptide (VnP-16) is a recently developed anti-osteoporotic agent [20]. VnP-16 enhances osteoblast differentiation via β1 integrin-FAK signaling and suppresses osteoclast differentiation and resorptive activity via JNK-c-Fos-NFATc1 and αvβ3 integrin-c-Src-PYK2 signaling, respectively [20]. In a mouse model of experimental autoimmune encephalitis, αv integrin expressed by dendritic cells was required for Th17 differentiation [21]. Also, integrin expressed by T cells is important for cell differentiation, migration, and costimulatory signaling [22]. However, the anti-inflammatory effect of VnP-16 has not been evaluated in an animal model of SpA.

Several SpA animal models have been introduced, including the curdlan-induced SKG mouse model [23]. SKG mice have the ZAP-70$^{W163C}$ mutation, and curdlan immunization

induces several SpA features, including ankylosis of the spine, gut inflammation, uveitis, and psoriasis-like skin lesions [23]. The ZAP-70$^{W163C}$ mutation plays a crucial role in thymic T cell selection; therefore, this SpA mouse model is useful for assessing not only structural damage of axial joints, but also T cell differentiation in the pathogenesis of SpA. The incidence of arthritis was higher in female SKG mice than in male SKG mice when immunized by curdlan [23]; accordingly, a recent study using the SKG mouse model preferred female over male SKG mice [24].

We evaluated the anti-arthritic effect of VnP-16 in mice with SpA. The clinical arthritis score, histologic severity grade, and proinflammatory cytokine expression in the spine were determined, and helper T-cell polarization to Th17 cells or Tregs was evaluated.

## Materials and methods

### Mice

Female SKG mice on a BALB/c background with ZAP-70$^{W163C}$ mutation and 8 to 10 weeks old were purchased from Saeronbio (Uiwang, South Korea). Mice were bred under specific-pathogen-free conditions and fed standard mouse chow (Ralston Purina, St. Louis, MO) and water *ad libitum*. The experiments were assessed and approved by the Institutional Animal Care and Use Committee of the School of Medicine and the Animal Research Ethics Committee of the Catholic University of Korea and were conducted in accordance with the Laboratory Animals Welfare Act and the Guide for the Care and Use of Laboratory Animals (no. CUMC-2019-0298-02).

### Induction of SpA and drug administration

Curdlan (3 mg/kg) was injected intraperitoneally (IP) into SKG mice aged 8–10 weeks. VnP-16 was synthesized as described [20], and the NSAID, celecoxib, was obtained from Hanlim Pharmaceutical. The mice were divided into four groups (n = 10 per group): 1) vehicle (phosphate-buffered saline) administration, 2) oral celecoxib 10 mg/day, 3) subcutaneous VnP-16 600 μg/week, and 4) oral celecoxib 10 mg/day with subcutaneous VnP-16 600 μg/week. The treatment started 1 week after curdlan injection and continued for 11 weeks. Following a previous study [23], clinical scores were measured weekly for 12 weeks by three independent experimenters; the mean (and standard error of the mean) scores were used in the analysis. The scores of the affected joints were summed for each mouse. The clinical score was assessed under isoflurane inhalation anesthesia in every mouse, and every effort was made to minimize suffering. After the mice had been euthanized with 100% carbon dioxide within 5 min, in accordance with the use and care of animal guidelines of the Catholic University of Korea, the joint, spleen, and spinal tissues were collected.

### Histopathological analysis

Tissue samples from the peripheral joints and spine were fixed in 10% neutral-buffered formalin, embedded in paraffin, and sectioned at a thickness of 7 μm. The sections were dewaxed using xylene, dehydrated in an alcohol gradient, stained with hematoxylin and eosin (H&E) and Safranin O, and were scored for inflammation. The histologic scores of the peripheral joints and spine were calculated as described previously [23]. Histopathological analysis was performed by three experimenters in a blinded fashion. Stained tissues were examined by photomicroscopy (Olympus, Tokyo, Japan, magnification 40×, 200×).

## Immunohistochemistry

Immunohistochemical analyses were performed using the Dako REAL™ Envision™ Detection System Kit (DAKO, Glostrup, Denmark, #5007). Tissues were first incubated with primary antibodies (Abs) against IL-1β IL-6, IL-17A, and tumor necrosis factor (TNF)-α (all from Abcam, Cambridge, UK) overnight at 4°C followed by incubation with Dako REAL™ Envision™/HRP for 30 min. The final colored product was developed using a chromogen diaminobenzidine. Three independent, blinded observers assessed all of the histologic scores. Images were taken using a DP71 digital camera (Olympus, Center Valley, PA, USA) attached to a BX41 microscope (Olympus). Positive cells were counted (magnification 400×) with the aid of Adobe Photoshop software and were averaged in three randomly selected fields per tissue section.

## Confocal microscopy

Spines were removed from mice at 12 weeks after curdlan injection. To assess the differentiation of Th17 cells and Tregs, the spine tissues were reacted with Abs against CD4–fluorescein isothiocyanate (FITC), IL-17–phycoerythrin (PE), CD25–allophycocyanin, and forkhead box P3 (Foxp3)–PE (all from eBioscience, San Diego, CA, USA). The stained tissue sections were visualized using a confocal microscope (LSM 700 Meta; Carl Zeiss, Oberkochen, Germany). Double or triple positive cells were counted in three high-power fields (magnification 400×) per section.

## Flow cytometric analyses

Cell pellets were prepared from spleen tissues. The regulatory T cell populations were examined using anti-mouse CD4–peridin chlorophyll protein (perCP) and anti-mouse CD25-allophycocyanin (APC) (eBioscience); then, the cells were fixed and permeabilized using a Foxp3/Transcription Factor Staining Buffer set (Thermo Fisher Scientific, Waltham, MA, USA) according to the manufacturer's instructions. For Th17 cell analysis, before FACs staining, the cells were stimulated with 25 ng/mL phosphomolybdic acid (Sigma-Aldrich, St. Louis, MO, USA), 250 ng/mL ionomycin (Sigma-Aldrich), and Golgi Stop (BD Biosciences, San Diego, CA, USA) in 5% $CO_2$ at 37°C for 4 hours. The cells were stained with anti-mouse CD4 PerCP, and then with an anti-mouse IL-17 FITC (eBioscience), followed by fixation and permeabilization using a Cytofix/Cytoperm Plus Kit (BD Biosciences) according to the manufacturer's instructions. The samples were analyzed using a FACSCalibur instrument (BD Pharmingen; BD Biosciences).

## Western blotting

The protein levels of *p*-STAT3(s727) (cat: #9134, 100kDa, Cell Signaling Technology, Beverly, MA, USA), total STAT3 (cat: #9189, 100 kDa, Cell Signaling Technology) and GAPDH (#ab181602; Abcam) were measured using a Western blot system (SNAP i.d. Protein Detection System; Merck Millipore, Danvers, MD, USA). Splenocytes were harvested from BALB/c mice and then stimulated with VnP-16 (100 μg/mL) or vehicle for 2 hours, and then with IL-6 (10 ng/mL) for 1 hour. Then, whole-cell lysates were prepared. The protein concentration was measured using the BCA assay method (#23235, Thermo), and samples were separated on a 4–12% sodium dodecyl sulfate polyacrylamide gel and transferred to a nitrocellulose membrane (Amersham Pharmacia, Uppsala, Sweden). The primary antibodies *p*-STAT3 s727 (cat: #9134, 100kDa; Cell Signaling Technology), total STAT3 (cat: #9189, 100 kDa; Cell Signaling Technology), and GAPDH (cat: ab181602, 36kDa; Abcam) were diluted in 0.1% skim milk in

Tris-buffered saline Tween-20 and incubated for 20 min at room temperature. The membrane was washed and incubated with horseradish peroxidase-conjugated secondary antibody for 20 min at room temperature. Band density was estimated by image capture densitometry.

## Statistical analyses

Continuous variables are presented as the mean ± standard error of the mean. Differences between groups were analyzed using the Kruskal–Wallis test. $P < 0.05$ was considered significant. The statistical analyses were performed using SPSS 20.0 for Windows (IBM Corp., Armonk, NY, USA).

## Results

### VnP-16 prevents SpA progression

The clinical scores were evaluated weekly (n = 10 per group) after curdlan injection. The clinical score was significantly attenuated in the VnP-16+celecoxib group from 4 weeks after curdlan injection to the time of euthanasia (Fig 1A). The arthritis scores of the VnP-16 alone and VnP-16+celecoxib groups were significantly lower than those of the vehicle group (Fig 1B). The arthritis score was also lower in the VnP-16+celecoxib than celecoxib-alone group (Fig 1B). The VnP-16 alone and VnP-16+celecoxib groups had lower spondylitis scores than the vehicle group (Fig 2). The difference in spondylitis score between the celecoxib alone and VnP-16+celecoxib groups was not significant. Immune cell infiltration and cartilage destruction in the spine were less severe in the VnP-16 alone and VnP-16+celecoxib groups than the vehicle group.

### VnP-16 suppressed inflammatory cytokine expression

The expression levels of IL-1β, IL-6, TNF-α, and IL-17, were assessed by IHC in the nucleus pulposus (NP) and cartilaginous end plate (CEP) of the vertebral corner. In the NP, IL-1β and TNF-α expression was significantly suppressed in the VnP-16+celecoxib group compared to the vehicle group, whereas IL-6 and IL-17A expression was significantly suppressed in the VnP-16 alone and VnP-16+celecoxib groups (Fig 3). Suppression of IL-17 was more marked in the VnP-16+celecoxib than celecoxib-alone group (Fig 3). In the CEP, the expression of the four inflammatory cytokines was suppressed in the VnP-16 alone and VnP-16+celecoxib groups (Fig 4). Furthermore, IL-1β and IL-6 expression were decreased in the VnP-16+celecoxib group compared to the celecoxib-alone group (Fig 4). Interestingly, celecoxib alone did not prevent disease progression, whereas VnP-16 alone significantly reduced the histology score of the peripheral joints/spine and proinflammatory cytokine expression in the spine. In addition, the combination of VnP-16+celecoxib suppressed inflammatory cytokine expression to a greater degree in the spine than celecoxib alone.

### Immunomodulatory role of VnP-16 in helper T-cell differentiation

In the annulus fibrosus area, CD4+ IL-17+ and CD4+ IL-22+ IL-17+ cells were downregulated, and CD4+ CD25+ Foxp3+ cells were upregulated, in the VnP-16 alone and VnP-16+celecoxib groups compared to the vehicle group (Fig 5A–5C). The decrease in CD4+ IL17+ cells and increase in CD4+ CD25+ Foxp3+ cells were significantly greater in the VnP-16+celecoxib group compared to the celecoxib-alone group (Fig 5A and 5C). Flow cytometry showed that CD4+ CD25+ Foxp3+ T cells were upregulated in the VnP-16+celecoxib group compared to the vehicle group (Fig 6). The raw flow cytometry data are provided as S1 Fig. The Th17 population (CD4+ IL-17+ T cells) tended to decrease in the VnP-16+celecoxib

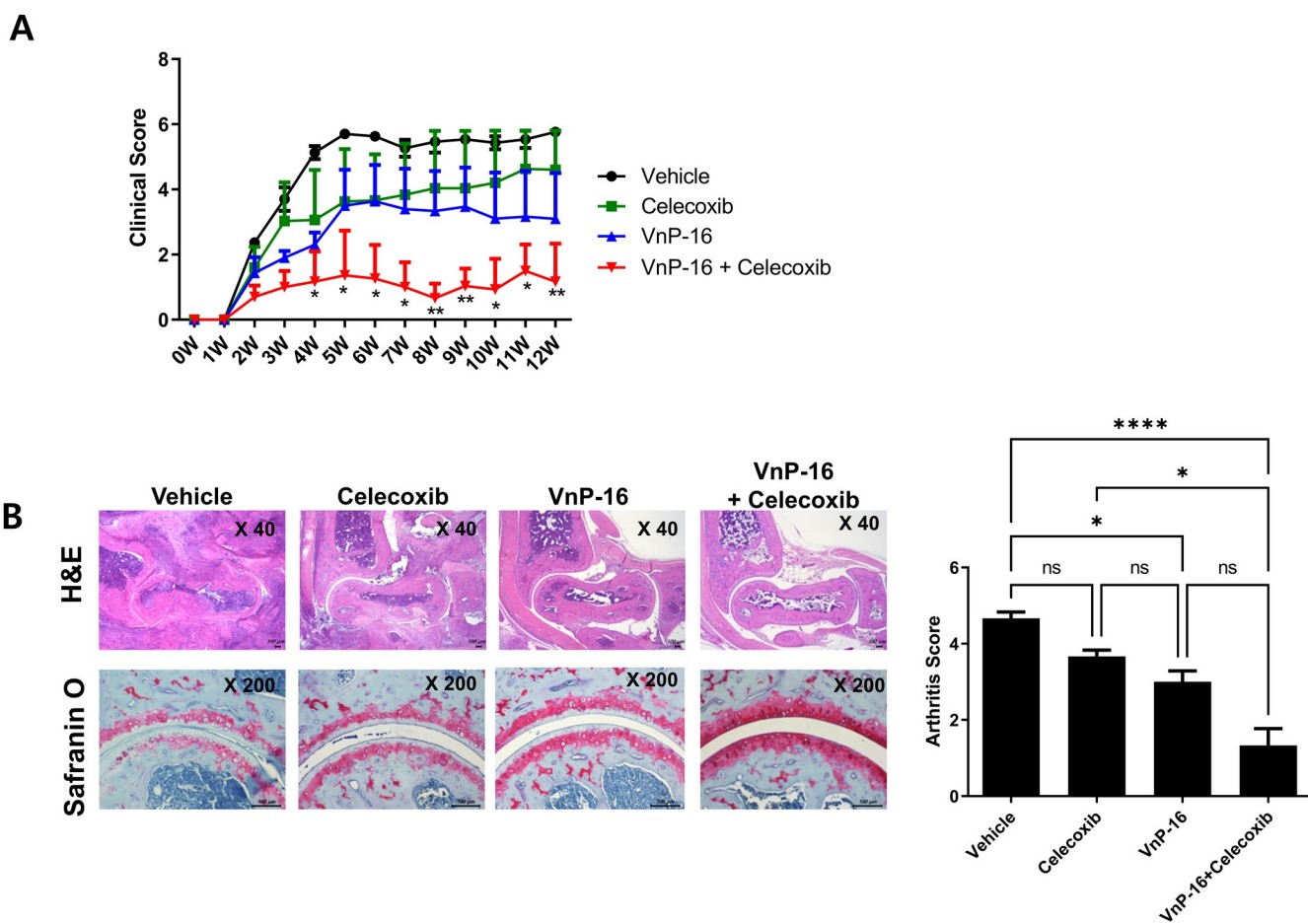

**Fig 1. Anti-arthritic effects of vitronectin-derived bioactive peptide (VnP-16) in spondyloarthritis (SpA) mice.** Curdlan (3 mg/kg) was injected intraperitoneally into SKG mice with the ZAP-70$^{W163C}$ mutation to induce SpA. The treatment groups were as follows: 1) vehicle (phosphate-buffered saline) administration; 2) oral celecoxib 10 mg/day; 3) subcutaneous VnP-16 600 µg/week; and 4) oral celecoxib 10 mg/day with subcutaneous VnP-16 600 µg/week. The *in vivo* experiments were repeated twice and pooled data are presented (n = 10 per group). (A) Weekly mean clinical score for 12 weeks. Black dot, vehicle group; green dot, celecoxib-alone group; blue dot, VnP-16 alone group; red dot, VnP-16+celecoxib group. (B) Hematoxylin and eosin (H&E)/ Safranin O-stained images of ankle joints isolated from SpA mice 12 weeks after curdlan injection; bar graphs show the arthritis score. Data are means ± standard error of the mean (SEM) of assessments by three independent experimenters. ns, non-significant; $^*P < 0.05$, $^{**}P < 0.01$, $^{****}P < 0.0001$.

compared to vehicle group. Furthermore, pSTAT3 s727 expression was significantly suppressed by VnP-16 100 µg/mL (Fig 7; raw data S2 Fig), which demonstrates the Th17/Treg regulation mechanism of VnP-16.

## Discussion

We evaluated the anti-arthritic effect of VnP-16 in mice with SpA. The clinical score, arthritis and spondylitis scores, and expression of proinflammatory cytokines in the spine were suppressed by VnP-16 plus celecoxib. VnP-16 alone reduced the arthritis and spondylitis scores and suppressed proinflammatory cytokine expression in the spine. Furthermore, VnP-16 alone and VnP-16+celecoxib regulated the Th17/Treg population in spine tissue, and VnP-16 +celecoxib augmented Treg differentiation in the spleen. The anti-arthritic effects of anti-osteoporotic agents could be useful in patients with inflammatory arthritis, including SpA, which are linked to increased osteoclast activity and an increased risk of osteoporosis [12]. The latter could be explained by increased systemic inflammation [15]; moreover, IL-1β, IL-6, IL-17, and

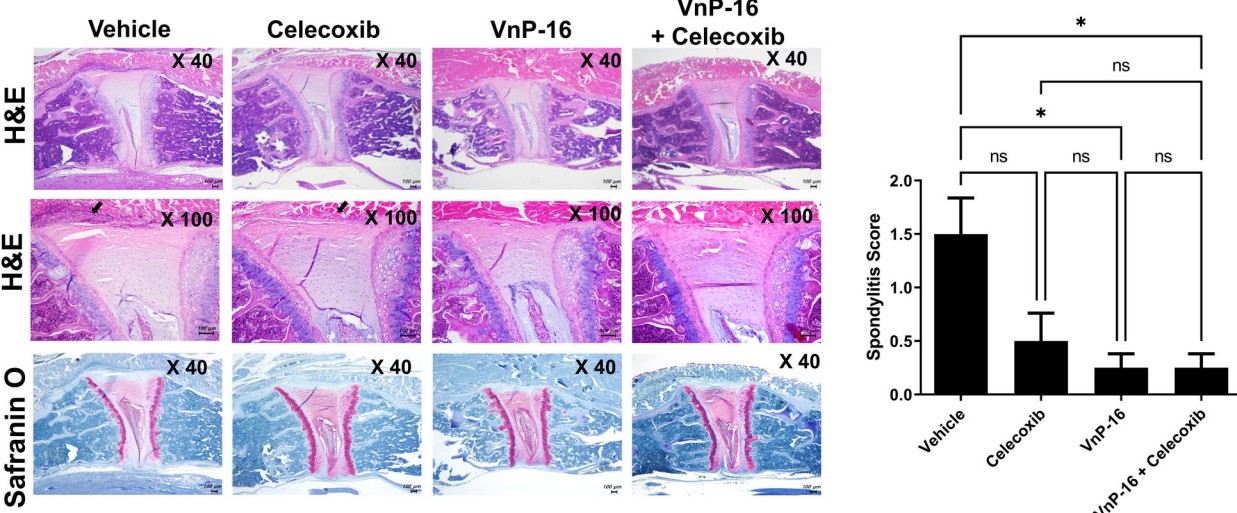

**Fig 2. VnP-16 reduces the spondylitis score of SpA mice.** *In vivo* experiments were repeated twice and pooled data are presented (n = 10 per group). Spine tissues were obtained from the vehicle-, celecoxib-, VnP-16 single-, and VnP-16+celecoxib-treated groups 12 weeks after curdlan injection and stained with H&E and Safranin O. Black arrows indicate inflammatory cell infiltration. Bar graphs show the spondylitis score. Ns, non-significant, *P < 0.05.

TNF-α promote osteoclastogenesis in inflammatory arthritis [25]. Therefore, additional anti-arthritic effects of anti-osteoporotic agent can be attractive therapy in patients with inflammatory arthritis, because it could expect dual therapeutic effects, preventing osteoporosis and additional anti-arthritic effects.

NSAIDs, including celecoxib, are used in patients with SpA as a first-line therapy [4]. Although NSAIDs ameliorate symptoms of SpA, such as arthralgia and stiffness, whether they can prevent spinal structural progression is unclear. Wanders et al. reported that NSAIDs suppress spinal structural progression in patients with AS [26]. However, several subsequent randomized and observational studies failed to reproduce the effect [27]. In the present study, NSAIDs alone did not suppress arthritis in SpA mice, but adding VnP-16 augmented the anti-arthritic and immunomodulatory effects by regulating the Th17/Treg balance and inflammatory cytokine expression. The effects on the arthritis score, inflammatory cytokine expression, and Th17/Treg imbalance were most prominent in the VnP-16+celecoxib treatment group. VnP-16 showed anti-osteoporotic effects by restraining the JNK-c-Fos-NFATc1 and αvβ3 integrin-c-Src-PYK2 signal pathways and promoting activity in the β1 integrin-FAK signal pathway [20]. Nuclear FAK induces Treg recruitment [28] and α4β1 integrin activation increases the immunosuppressive capacity of Tregs [29]. The inflammatory response of Th17 cells in a mouse model of multiple sclerosis depended on αvβ3 integrin signaling [30], and NFATc1 was required to induce the Th17 transcription factor RORγt [31]. Therefore, VnP-16 may regulate Th17/Tregs via these signaling pathways. Although the precise mechanism of the potential synergistic effect of VnP-16 and celecoxib was not uncovered here, the results suggest additive therapeutic effects of VnP-16 on SpA. The second-line therapy for SpA is TNF-α inhibitors [4], long-term treatment with which attenuated spinal structural progression in patients with SpA [32]. In addition, the reduction by TNF-α inhibitors of spinal structural damage has been suggested to be mediated by suppression of inflammation and C-reactive protein (CRP) [33, 34], and the intrinsic effects of TNF-α inhibitors (which are independent of inflammation and CRP control) [35]. IL-6 stimulates the production of CRP by hepatocytes [36]. Therefore, controlling the expression of proinflammatory cytokines has therapeutic

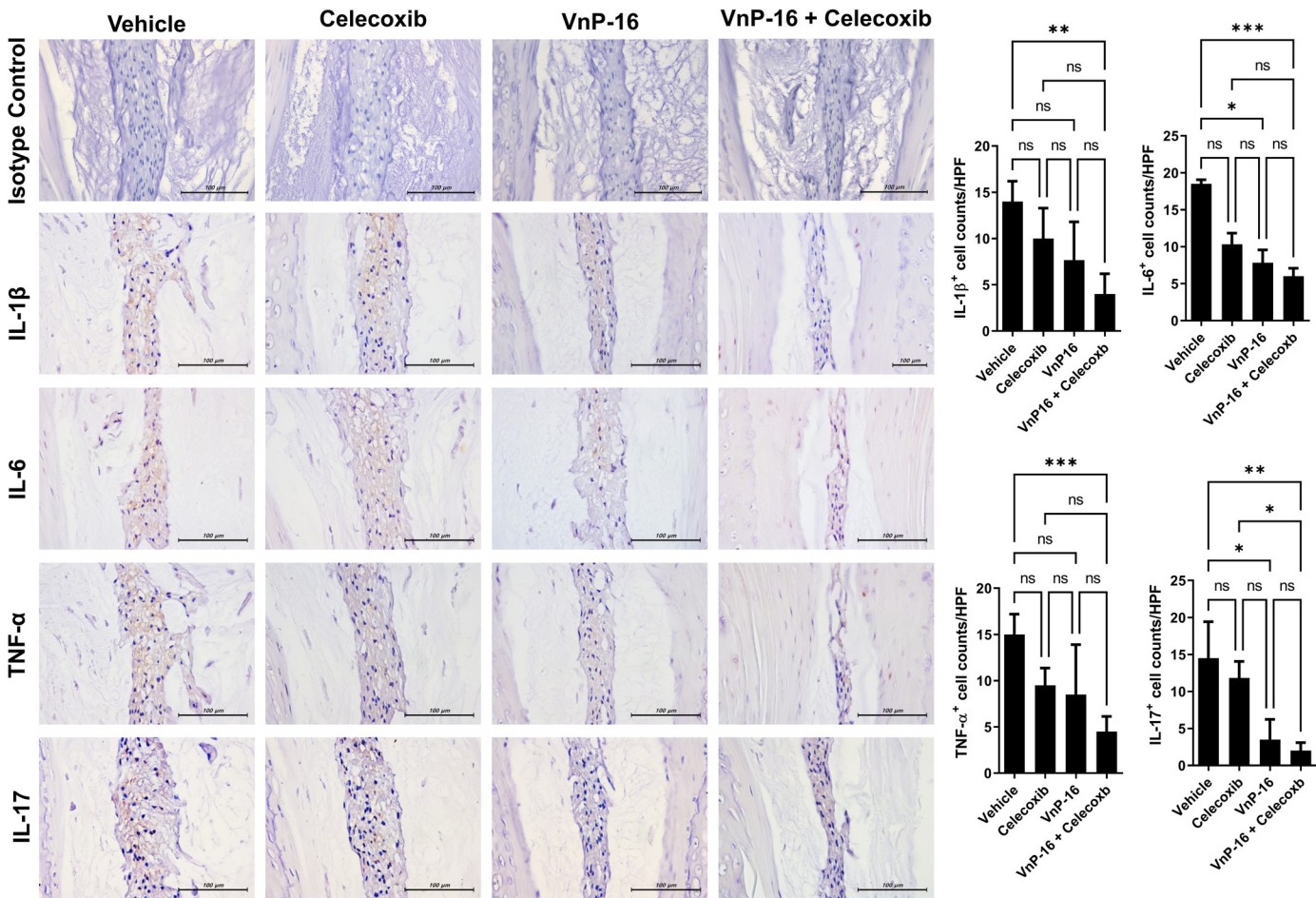

**Fig 3. VnP-16 reduces inflammatory cytokine expression in the nucleus pulposus of mice with SpA.** IL-1β, IL-6, TNF-α, IL-17A expressing cells were enumerated in the nucleus pulposus by immunohistochemical staining (n = 6 per group). ns, non-significant, *P < 0.05, **P < 0.01, ***P < 0.001.

potential for SpA. In this study, VnP-16 alone suppressed proinflammatory cytokine (TNF-α and IL-6) expression in the spine. Therefore, the therapeutic effect of VnP-16 on spinal structural progression in SpA is mediated by modulation of proinflammatory cytokine expression.

Although the pathogenesis of SpA is unclear, helper T cells are implicated. Th17 cells produce IL-17 and they are more numerous in patients with SpA; Th17 cell expression correlated with SpA disease activity [7, 8]. Furthermore, IL-22+ expressing Th17 cells played an important role in joint destruction in a mouse model of inflammatory arthritis [37]. Tregs promote immune tolerance and attenuate inflammatory responses in various immune-mediated diseases [38]. STAT3 inhibitor regulated the Th17/Treg imbalance in the SpA mouse model [10]. SKG mice have a rheumatoid arthritis-like phenotype mediated by spontaneous Th17 polarization [39]. In addition, transplantation of CD4+ T cells extracted from curdlan-induced SpA mice into severe combined immunodeficient mice induced features of SpA [23]. Therefore, the SKG mouse enables assessment of the immunoregulatory mechanism of novel medications in terms of the T-cell-mediated response. In this study, Treg expression was increased in both the spine and spleen, while Th17 and IL-22+ Th17 was significantly decreased in spine tissue of SpA mice by VnP-16. The Th17 population in the spleen also tended to decrease with VnP-16 treatment. Although the suppression of Th17 by VnP-16 in the spleen was not significant,

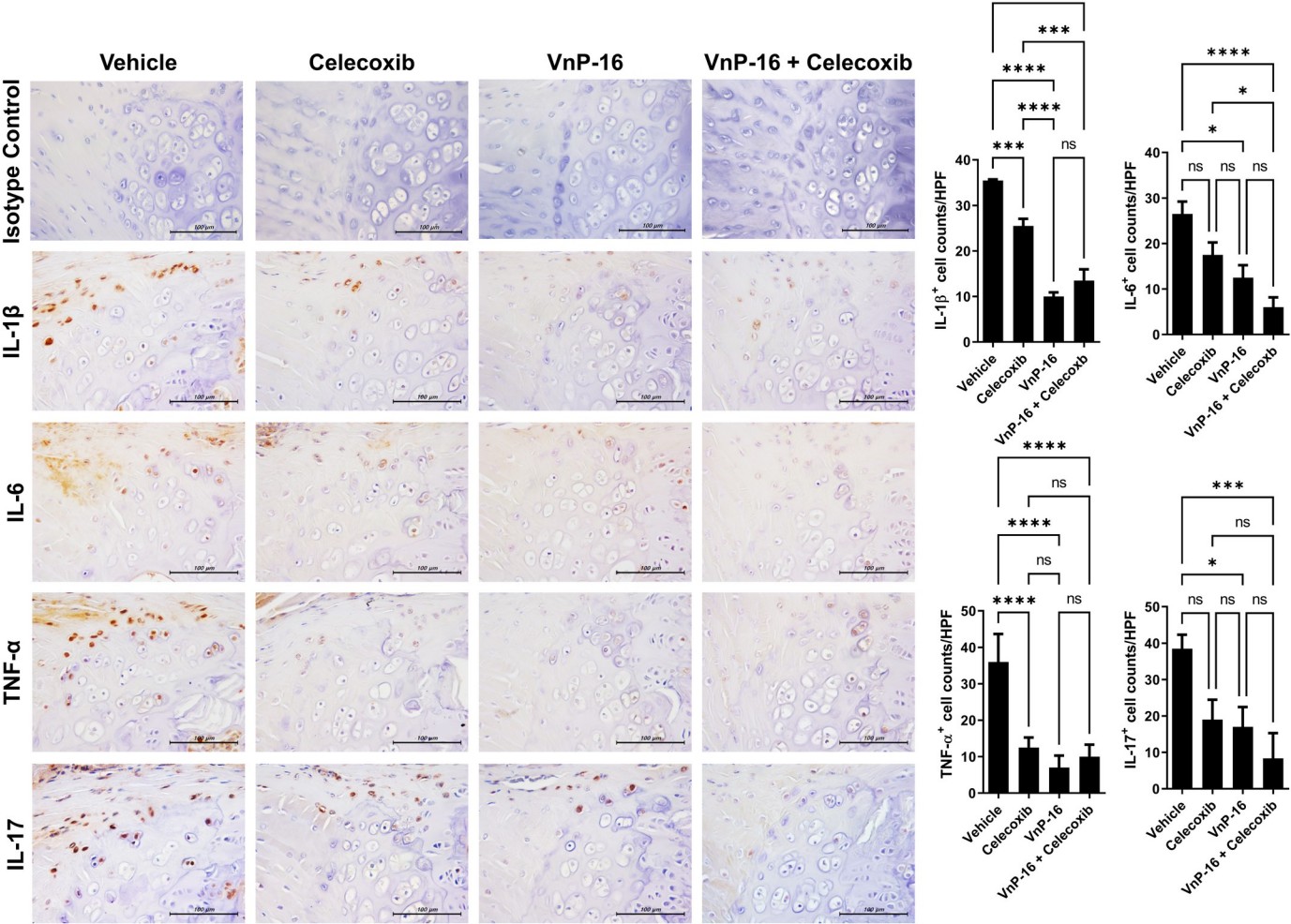

**Fig 4. VnP-16 reduces inflammatory cytokine expression in the cartilaginous end plate of mice with SpA.** Immunohistochemical staining of IL-1β, IL-6, TNF-α, and IL-17A in the cartilaginous end plate (n = 6 per group). Data are means ± SEM. ns, non-significant, *$P < 0.05$, ***$P < 0.001$, ****$P < 0.0001$.

*p*STAT3 regulation by VnP-16 was confirmed *in vitro*. These results suggest a regulatory role of VnP-16 on the Th17/Treg imbalance via regulation of STAT3 in the SpA mouse model; this is one of the anti-arthritic mechanisms mediated by VnP-16.

Prevention of syndesmophyte formation is important, because it is irreversible and can cause substantial limitation of motion and reduce the quality of life of patients with SpA. Abnormal new bone formation involves the following steps: 1) acute inflammation in the vertebral corner, presenting as bone marrow edema by magnetic resonance imaging (MRI), 2) chronic changes that appear as fat metaplasia by MRI, and 3) formation of syndesmophytes and a bony bridge at the vertebral corner [40]. Bone marrow edema (acute inflammation) precedes fat metaplasia, which increases the risk of syndesmophyte formation [41]. Therefore, proper management at the acute inflammation stage (bone marrow edema) can prevent fat metaplasia and syndesmophyte formation in patients with SpA [41, 42]. In addition, bone biopsy of bone marrow edema showed osteoclast predominance, whereas osteoblasts were predominant in fat metaplasia [43]. Bone-derived cells from the facet joints of patients with SpA showed increased osteoblast differentiation upon stimulation with IL-17, suggesting that IL-17 is crucial for abnormal new bone formation in SpA [44]. In this study, VnP-16 reduced IL-17

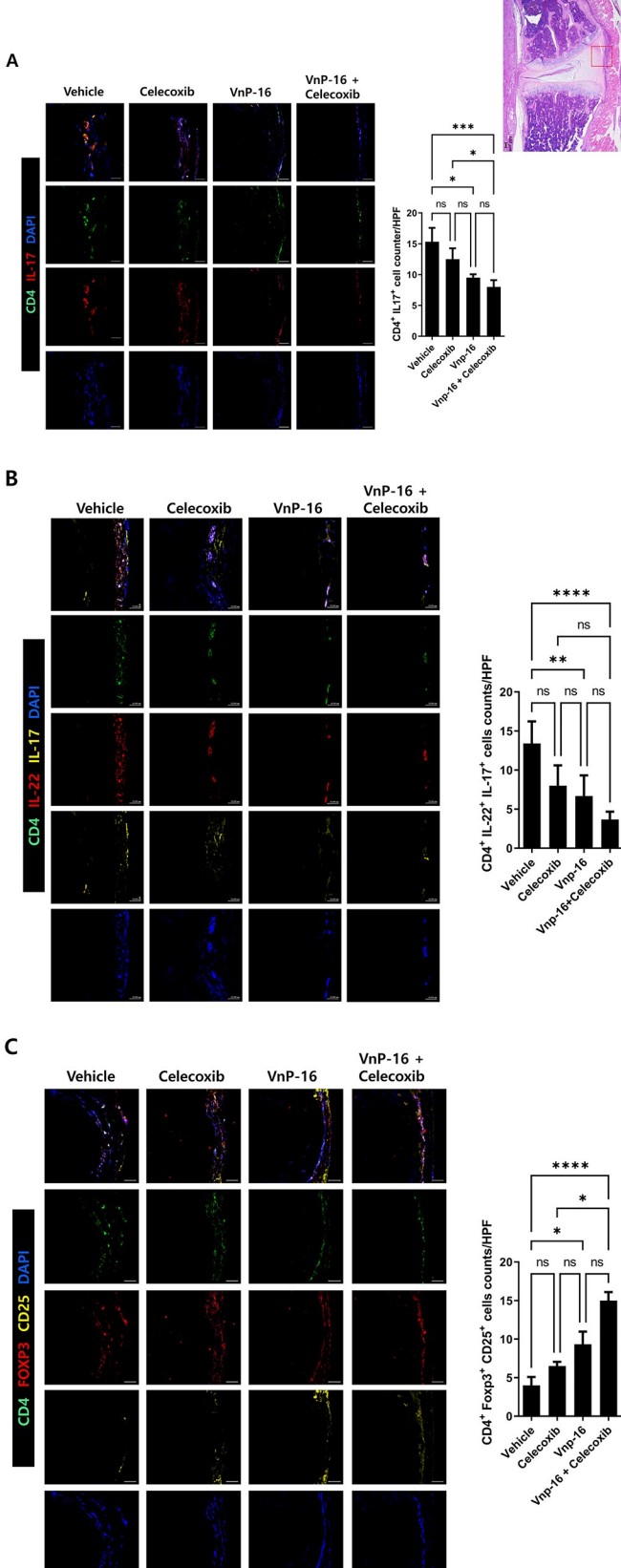

**Fig 5. VnP-16 regulates type 17 helper T cell (Th17) and regulatory T cell (Treg) populations in the annulus fibrosus of SpA mice.** Spine tissue was stained with CD4–FITC, IL-17–PE, CD25–APC, Foxp3–PE to evaluate (A) Th17, (B) IL-22+ Th17, and (C) Treg populations (n = 6 per group). Double-positive cells are shown in the bar graph. Data are means ± SEM. ns, non-significant, *$P < 0.05$, **$P < 0.01$, ***$P < 0.001$, ****$P < 0.0001$.

expression in the spine of SpA mice. Considering its originally documented action, i.e., osteoclast inhibition [20], VnP-16 has potential to suppress new bone formation by suppressing the expression of the osteoblast-activating cytokine IL-17 and inhibiting osteoclastogenesis at an early stage (bone marrow edema). However, further studies are required to reveal the effects of VnP-16 on syndesmophyte formation in SpA.

VnP-16 inhibited αvβ3 integrin-c-Src-PYK2–mediated bone resorption and enhanced β1 integrin-FAK signaling, promoting osteoblast differentiation [20]. Integrin expressed on T cells promotes interactions with neighboring cells, cytoskeletal organization, and migration. Integrins comprise α and β subunits, which have different effects on T cells. LFA-1 (αLβ2 integrin) and VLA-4 (α4β1 integrin) act on T-cell differentiation, extravasation, and costimulatory signaling, whereas α4β7 integrin interacts with MAdCAM-1 to induce gut homing of T cells [22, 45]. VnP-16 may have immunoregulatory effects on cells of the monocyte/macrophage lineage or integrin-mediated signaling. Further studies are needed to confirm the mechanism.

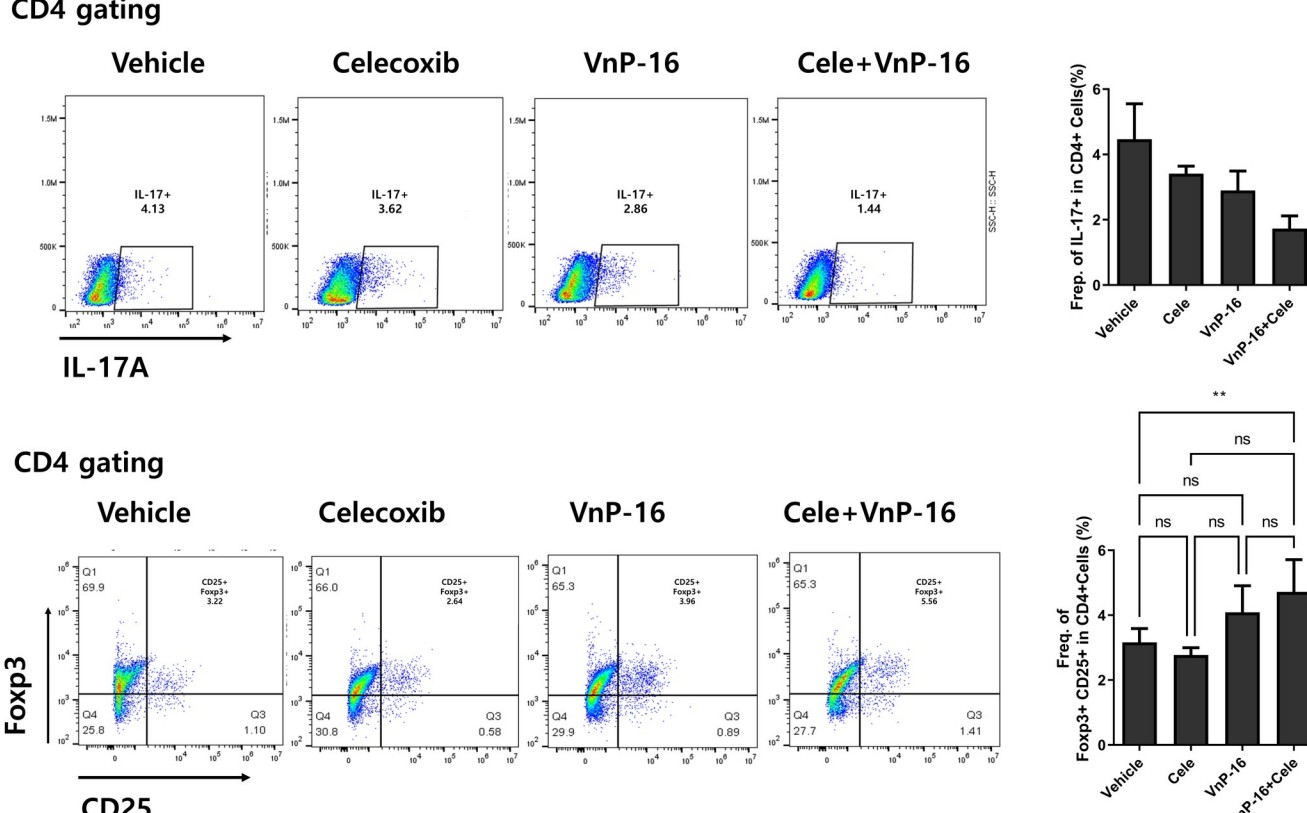

**Fig 6. VnP-16 regulates type 17 helper T cell (Th17) and regulatory T cell (Treg) differentiation in the spleens of SpA mice.** Splenocytes were subjected to flow cytometry using antibodies against IL-17A, CD4, CD25, and Foxp3 to determine the Th17 and Treg populations (n = 3 per group). Data are means ± SEM. ns, non-significant, **$P < 0.01$.

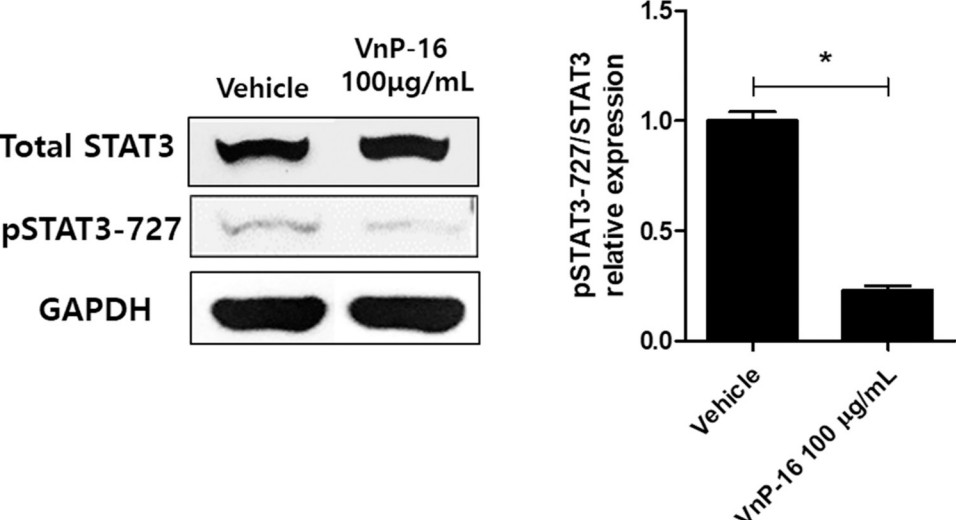

**Fig 7. VnP-16 suppresses pSTAT3 s727 expression of splenocytes.** Splenocytes harvested from BALB/c mice were stimulated with vehicle or VnP-16, and total STAT3, $p$STAT3 s727, and GAPDH expression were measured by Western blotting (n = 3 per group). Data are means ± SEM. *$P < 0.05$. $P$-values are in comparison with the vehicle group.

## Conclusions

In conclusion, VnP-16 showed an anti-arthritic effect in SpA mice by modulating the Th17/ Treg imbalance and suppressing inflammatory cytokine expression in axial joints. VnP-16 plus an NSAID prevented SpA and ameliorated peripheral arthritis and spondylitis. Therefore, VnP-16 exerts a protective effect against SpA.

## Supporting information

**S1 Fig. Raw data for the Th17 and Treg flow cytometric results for splenocytes.**
(TIF)

**S2 Fig. Raw Western blot data.**
(TIF)

## Author Contributions

**Conceptualization:** Hong Ki Min, Byung-Moo Min, Mi-La Cho, Sung-Hwan Park.

**Data curation:** Hong Ki Min, JeongWon Choi, Seon-Yeong Lee, A. Ram Lee.

**Formal analysis:** Hong Ki Min, JeongWon Choi.

**Funding acquisition:** Sung-Hwan Park.

**Investigation:** JeongWon Choi, Seon-Yeong Lee.

**Methodology:** Hong Ki Min, Mi-La Cho.

**Resources:** Byung-Moo Min.

**Supervision:** Byung-Moo Min, Mi-La Cho, Sung-Hwan Park.

**Writing – original draft:** Hong Ki Min.

**Writing – review & editing:** Byung-Moo Min, Mi-La Cho, Sung-Hwan Park.

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
