## [Decision Letter · Decision Letter 0]

15 Nov 2021

PONE-D-21-32711Vitronectin-derived bioactive peptide prevents spondyloarthritis by modulating Th17/Treg imbalance in mice with curdlan-induced spondyloarthritisPLOS ONE

Dear Dr. Park,

Thank you for submitting your manuscript to PLOS ONE. After careful consideration, we feel that it has merit but does not fully meet PLOS ONE’s publication criteria as it currently stands. Therefore, we invite you to submit a revised version of the manuscript that addresses the points raised during the review process. Two experts in T cell immunity have reviewed the manuscript and raised a number of concerns that need to be addressed in the revised manuscript. In particular, both reviewers suggested that the authors can improve the flow data in Fig 6 & 7. In addition, the authors are also encouraged to improve the description of their data and to expand discussion on the potential mechanisms.

We look forward to receiving your revised manuscript.

Kind regards,

Yeonseok Chung

Academic Editor

PLOS ONE

Journal Requirements:

Reviewers' comments:

Reviewer's Responses to Questions

**Comments to the Author**

1. Is the manuscript technically sound, and do the data support the conclusions?

Reviewer #1: Partly

Reviewer #2: Yes

2. Has the statistical analysis been performed appropriately and rigorously? 

Reviewer #1: No

Reviewer #2: Yes

3. Have the authors made all data underlying the findings in their manuscript fully available?

Reviewer #1: Yes

Reviewer #2: Yes

4. Is the manuscript presented in an intelligible fashion and written in standard English?

Reviewer #1: No

Reviewer #2: Yes

5. Review Comments to the Author

Reviewer #1: In this manuscript, Min et al. reported that vitronectin-derived peptide, VnP16, ameliorates spondyloarthritis (SpA) in SKG mice treated with curdlan. The authors firstly showed that either VnP16 or VnP16+celecoxb attenuated inflammation at ankle joints and spines of SpA-induced mice compared to that of vehicle-treated mice. Furthermore, VnP16 treatment in SpA-induced mice exhibited potent inhibitory effect on the production of IL-1β, IL-6, TNFα, and IL-17 in nucleus pulposus and cartilaginous. Fluorescence microscopic analyses of SpA-induced mice revealed the reduction of Th17/Treg ratio upon VnP16 treatment.

Overall, this manuscript presented novel effects of VnP16 on Th17/Treg balance during SpA. Addressing the following minor points will strengthen the clarity of this study.

Minor points

1. In the Legends, the authors should describe which mouse strain/chemical they used to induce SpA along with treatment schedule.

2. Regarding following statement in the Legend of Figure 1: “Data are means ± standard error of the mean (SEM) of three independent experiments.”, it is confusing if those data are pooled from 3 independent experiments or representative of 3 independent experiments.

3. To solidify their conclusion, authors should present more statistical analysis through the overall experimental groups, not just vehicle vs VnP16 or VnP16+celecoxib.

4. In the Legend of Figure 6, the authors appear to analyze splenocytes, not spleen tissue itself.

5. Flow cytometric analyses of Th17 and Treg cells in Figure 7 look ambiguous in terms of staining and gating. Since not all cellular responses in tissues are reconciled systemically, it would be better to exclude Figure 7 if the authors are unable to show more clear data on splenocytes.

6. On page 13, please provide a reference for the following statement: “Therefore, proper management at the acute inflammation stage (bone marrow edema) can prevent fat metaplasia and syndesmophyte formation in patients with SpA.”

7. The manuscript requires editing to conform to correct scientific English.

Reviewer #2: In this manuscript, the authors investigated the therapeutic potential of Vitronectin-derived bioactive peptide in combination with anti-inflammatory drug celecoxib for the treatment of spondyloarthritis. The combination between the two drugs was more effective than each monotherapy for ameliorating the arthritis and spondylitis symptoms in an animal model of spondyloarthritis by diverting the balance between Th17 cells and Treg cells toward increased Treg/Th17 ratio.

Here are some specific comments.

1. Flow cytometric analysis shown in Figure 6 should be improved or removed. Gated IL-17+ and CD25+Foxp3+ cell populations in the Figure might be cellular debris or noise.

2. Please discuss the potential mechanism for the additive effect between VnP-16 and celecoxib in the regulation of Th17/Treg regulation in SpA.

3. Please describe the data in the Figures in more detail in the Result section.

4. Please indicate the number of experimental repetitions in each figure.

6. PLOS authors have the option to publish the peer review history of their article (what does this mean?). If published, this will include your full peer review and any attached files.

Reviewer #1: No

Reviewer #2: No

---

## [Author Response · Author response to Decision Letter 0]

9 Dec 2021

Revised submission of manuscript PONE-D-21-32711

Dear Pf. Yeonseok Chung

We thank the reviewers for their constructive and helpful comments concerning the manuscript. We have addressed the reviewers' concerns by revising the manuscript or explaining respectfully our rebuttal. The point-by-point replies are given in this letter. We hope that we have addressed satisfactorily all concerns raised by the reviewers, and that this manuscript is now suitable for publication.

Thank you again for the comments.

Sincerely yours,

Mi-La Cho, PhD

Conversant Research Consortium in Immunologic disease, College of Medicine, The Catholic University of Korea, Seoul, 222 Banpo-Daero, Seocho-gu, Seoul 06591, Republic of Korea 

Tel: +82-2-2258-7467, Fax: +82-2-599-4287, E-mail: iammila@catholic.ac.kr

Sung-Hwan Park, MD, PhD, Division of Rheumatology, Department of Internal Medicine, College of Medicine, Seoul St. Mary's Hospital, The Catholic University of Korea, 222 Banpo-Daero, Seocho-gu, Seoul 06591, Republic of Korea

Tel: +82-2-2258-6011, Fax: +82-2-599-3589, E-mail address: rapark@catholic.ac.kr

 

Journal Requirements:

Answer : We followed the PLOS ONE's style requirement.

Answer : We added the methods for sacrifice / anesthesia / effort to reduce suffering in method section (revised manuscript, line 146-150)

“The clinical score was assessed under isoflurane inhalation anesthesia in every mouse, and every effort was made to minimize suffering. After the mice had been euthanized with 100% carbon dioxide within 5 min, in accordance with the use and care of animal guidelines of the Catholic University of Korea, the joint, spleen, and spinal tissues were collected.” 

Answer : We added raw flow cytometry and western blot data of Figure 6 and 7 as supplementary figure 1 and 2 (revised manuscript, line 260, 263).

Answer : We added raw blot / gel image as supplementary figures 1 &2 and, and mentioned it in cover letter.

Answer : We mentioned the change in reference in rebuttal letter and marked by "track change” in “revised manuscript with track change” file.

 

Reviewers' comments:

Reviewer #1: In this manuscript, Min et al. reported that vitronectin-derived peptide, VnP16, ameliorates spondyloarthritis (SpA) in SKG mice treated with curdlan. The authors firstly showed that either VnP16 or VnP16+celecoxb attenuated inflammation at ankle joints and spines of SpA-induced mice compared to that of vehicle-treated mice. Furthermore, VnP16 treatment in SpA-induced mice exhibited potent inhibitory effect on the production of IL-1β, IL-6, TNFα, and IL-17 in nucleus pulposus and cartilaginous. Fluorescence microscopic analyses of SpA-induced mice revealed the reduction of Th17/Treg ratio upon VnP16 treatment.

Overall, this manuscript presented novel effects of VnP16 on Th17/Treg balance during SpA. Addressing the following minor points will strengthen the clarity of this study.

Minor points

1. In the Legends, the authors should describe which mouse strain/chemical they used to induce SpA along with treatment schedule.

Answer: We have added information on the mouse strain and chemicals used to induce SpA, and the treatment schedule, to the figure legends (revised manuscript, lines 559–563).

2. Regarding following statement in the Legend of Figure 1: “Data are means ± standard error of the mean (SEM) of three independent experiments.”, it is confusing if those data are pooled from 3 independent experiments or representative of 3 independent experiments.

Answer: We apologize for the typo. The clinical score and histological analysis were assessed by three independent experimenters blinded to the treatments. We calculated the mean values for the three experimenters. We have revised the figure legends (revised manuscript, lines 143–145, 568-569).

3. To solidify their conclusion, authors should present more statistical analysis through the overall experimental groups, not just vehicle vs VnP16 or VnP16+celecoxib.

Answer: We agree. We have added a comparison of the groups to all figures and the revised Results section (revised manuscript, lines 228–229, 230–232, 241–242, 243–245, 247–249, and 255–258)

4. In the Legend of Figure 6, the authors appear to analyze splenocytes, not spleen tissue itself.

Answer: We apologize for the typos. We have revised the legend of Figure 6 (line 597)

5. Flow cytometric analyses of Th17 and Treg cells in Figure 7 look ambiguous in terms of staining and gating. Since not all cellular responses in tissues are reconciled systemically, it would be better to exclude Figure 7 if the authors are unable to show more clear data on splenocytes.

Answer: We have improved the flow cytometry image in Figure 6.

6. On page 13, please provide a reference for the following statement: “Therefore, proper management at the acute inflammation stage (bone marrow edema) can prevent fat metaplasia and syndesmophyte formation in patients with SpA.”

Answer: We have cited a reference for this and have added reference number 42 (line 334).

7. The manuscript requires editing to conform to correct scientific English.

Answer: We have had the manuscript proofread by English experts. We have added a certificate pertaining to the English editing.

The English in this document has been checked by at least two professional editors, both native speakers of English. For a certificate, please see:

http://www.textcheck.com/certificate/pqmY6N

 

Reviewer #2: In this manuscript, the authors investigated the therapeutic potential of Vitronectin-derived bioactive peptide in combination with anti-inflammatory drug celecoxib for the treatment of spondyloarthritis. The combination between the two drugs was more effective than each monotherapy for ameliorating the arthritis and spondylitis symptoms in an animal model of spondyloarthritis by diverting the balance between Th17 cells and Treg cells toward increased Treg/Th17 ratio.

Here are some specific comments.

1. Flow cytometric analysis shown in Figure 6 should be improved or removed. Gated IL-17+ and CD25+Foxp3+ cell populations in the Figure might be cellular debris or noise.

Answer: We have improved the flow cytometry image in Figure 6.

2. Please discuss the potential mechanism for the additive effect between VnP-16 and celecoxib in the regulation of Th17/Treg regulation in SpA.

Answer: We agree. We have added a potential mechanism to the Discussion (revised manuscript, lines 289–296), and have also added references 28 to 31.

3. Please describe the data in the Figures in more detail in the Result section.

Answer: We have added more detailed descriptions of the figures to the Results (revised manuscript, lines 228–229, 230–232, 241–242, 243–245, 247–249, and 255–258).

4. Please indicate the number of experimental repetitions in each figure.

Answer: We have indicated the number of replicate experiments or total number of samples included in the analysis to each figure legend (revised manuscript, lines 563-564, 572–573, 581, 586, 592, 599, and 604).

---

## [Editor Report · Decision Letter 1]

19 Dec 2021

Vitronectin-derived bioactive peptide prevents spondyloarthritis by modulating Th17/Treg imbalance in mice with curdlan-induced spondyloarthritis

PONE-D-21-32711R1

Dear Dr. Park,

We’re pleased to inform you that your manuscript has been judged scientifically suitable for publication and will be formally accepted for publication once it meets all outstanding technical requirements.

Kind regards,

Yeonseok Chung

Academic Editor

PLOS ONE

Additional Editor Comments (optional):

The authors adequately addressed all critiques raised by the reviewers
---

## [Editor Report · Acceptance letter]

24 Dec 2021

PONE-D-21-32711R1 

Vitronectin-derived bioactive peptide prevents spondyloarthritis by modulating Th17/Treg imbalance in mice with curdlan-induced spondyloarthritis 

Dear Dr. Park:

I'm pleased to inform you that your manuscript has been deemed suitable for publication in PLOS ONE. Congratulations! Your manuscript is now with our production department. 

Kind regards, 

on behalf of

Dr Yeonseok Chung 

Academic Editor

PLOS ONE